# Eugenol, Isoeugenol, Thymol, Carvacrol, and Ester Derivatives as an Ecofriendly Option to Control Glomerella Leaf Spot and Bitter Rot on Apple

**DOI:** 10.3390/plants13223196

**Published:** 2024-11-14

**Authors:** Renan R. Schorr, Meira J. Ballesteros Garcia, Debora Petermann, Rafaele R. Moreira, Beatriz H. L. N. Sales Maia, Francisco A. Marques, Louise L. May-De Mio

**Affiliations:** 1Laboratório de Ecologia Química e Síntese de Produtos Naturais (LECOSIN), Departamento de Química, Universidade Federal do Paraná (UFPR), Av. Cel. Francisco H. dos Santos, 100 Jardim das Américas, Curitiba 81531-980, Brazil; renanreisschorr@hotmail.com (R.R.S.); meira.ballesteros@gmail.com (M.J.B.G.); bhsalesmaia@gmail.com (B.H.L.N.S.M.); fassismarques@yahoo.com.br (F.A.M.); 2Laboratório de Epidemiologia para Manejo Integrado de Doenças de Plantas (LEMID), Departamento de Fitotecnia e Fitossanidade, Universidade Federal do Paraná, Rua dos Funcionários, 1540, Curitiba 80035-050, Brazil; debora_peter@hotmail.com (D.P.); rafaelemor@gmail.com (R.R.M.)

**Keywords:** antifungal activity, natural substances, disease management, *Colletotrichum nymphaeae*, *Colletotrichum chrysophillum*

## Abstract

Glomerella leaf spot (GLS) and bitter rot (BR) are severe diseases of apple. *Colletotrichum nymphaeae* and *Colletotrichum chrysophillum* are the main species in Brazil. To control GLS and BR in Brazilian apple orchards, mancozeb and thiophanate-methyl fungicides are still used despite reported *Colletotrichum* resistance to these active ingredients. In addition, mancozeb has been banned from apple-importing countries and it has been a great challenge for apple producers to find products for its replacement that are eco-friendly. So, this study aimed to search for alternatives to control the diseases. We assessed the antifungal activity of eugenol, isoeugenol, thymol, carvacrol, and some of their ester derivatives. The best products to inhibit the pathogen in in vitro assays were thymol, thymol butyrate, and carvacrol, completely inhibiting mycelial growth at 125 mg L^−1^ and conidial germination at 100 mg L^−1^. In detached apple fruit, eugenol, eugenyl acetate, carvacryl acetate, and thymol butyrate, significantly reduced BR symptoms caused by *Colletotrichum* species with some variation between experiments and species, decreasing the risk of BR with the time compared to control. In detached leaves, all tested compounds significantly reduced the risk of development of GLS symptoms with disease control varying from 30 to 100%. The compounds tested are promising alternatives to replace fungicides to control bitter rot and Glomerella leaf spot on apple culture and should be tested for field conditions.

## 1. Introduction

The apple is a temperate fruit that has been cultivated in Europe and Asia since ancient times. Its genetic variability enabled the adaptation of apple trees to different environments. Currently, apples are distributed in almost all regions of the world and are consumed and market in many ways. Apples are one of the most produced fruits in the world, reaching about 10% of the produced fruits in the world in 2019 [1]. The global Apple fruit production reached 93,144,358 tons in 2021 [2], and China was the largest producer of this fruit, with 45,983,400 tons [2]. In Brazil, the production was 983,247 tons in 2020 [2].

The productivity of apple orchards faces significant challenges due to disease occurrences, notably Glomerella Leaf Spot (GLS) and Bitter Rot (BR), recognized as among the most severe diseases affecting apple yield worldwide [3,4]. In Brazil, GLS and BR hold importance due to their impact on the ‘Gala’ cultivar and its clones, which represent the most extensively cultivated varieties in the country [5,6]. While various *Colletotrichum* species have been linked to GLS and BR [7,8], recent studies have identified additional species of the same genus associated with these diseases, both in Brazil and globally [7,8,9].

Chemical intervention remains crucial in combatting this pathogen within commercial orchards [10,11]. In Brazil, numerous fungicides with active ingredients across distinct chemical groups are officially registered with the Brazilian Ministry of Agriculture for managing *Colletotrichum* species in apple orchards. Predominant among these fungicides are dithiocarbamates (including mancozeb, propineb, and metiram), methyl benzimidazole carbamates (such as carbendazim and thiophanate-methyl), as well as inorganic compounds like cuprous oxide, among others [12].

The continuous use of fungicides for disease control in apple trees can select resistant fungal isolates [13,14], which is one of the main problems of chemical control [15]. Fungicides currently available on the market pose an imminent risk of direct exposure to humans when used incorrectly, in addition to restrictions on the use of many of these products due to their carcinogenicity, teratogenicity, high and acute residual toxicity, long periods required for degradation, environmental pollution, food effects, and other side effects in humans [16].

Several studies have highlighted the emergence of resistance among apple pathogens towards primary active compounds utilized for disease management in this crop. Investigation by Moreira et al. (2017) [17] revealed that, among 39 *Colletotrichum* spp. isolates exposed to varying concentrations of the fungicides mancozeb and thiophanate-methyl in vitro, 21.4% exhibited resistance or high resistance, with 35.7% demonstrating moderate resistance to mancozeb. Isolates with phenotype of resistance or high resistance to thiophanate-methyl reached 73.6% of the sample. Similar findings have been reported by other researchers examining the sensitivity of *Colletotrichum* species from diverse fruit trees to various fungicides, including dithiocarbamates [18,19,20,21]. The proliferation of these resistant strains has resulted in field control failures, leaving producers with limited alternatives for managing GLS and BR effectively [10].

In this scenario, exploring alternative strategies for disease management becomes important, particularly in developing novel fungicidal options that exert minimal impact on human health and the environment. Global literature underscores biological control as a promising avenue, aiming to supplant or diminish the use of pesticides hazardous to humans and the ecosystem [22,23,24,25]. Biological control mechanisms have been extensively explored for *Colletotrichum* spp. causing diseases on several hosts [26,27,28,29]. Despite Brazil’s subtropical conditions favoring disease dissemination, implementing biological control measures for apple diseases under field conditions has proven challenging [30].

The use of essential oils and their major components has also been an option in the replacement of toxic pesticides. Many of these substances proven efficacy in food preservation, preventing the proliferation of organisms that damage food quality and thus compromising health safety [31,32,33]. Similarly, some studies have shown the effectiveness of these oils in controlling phytopathogenic organisms [34,35,36,37,38], thus being considered a promising alternative in plant diseases control.

Some natural compounds as eugenol, isoeugenol, thymol and carvacrol are known to present anti-inflammatory, antioxidant, antibacterial, antifungal and anticarcinogenic activity [16,39,40,41]. These biological activities also extend to the effects that these compounds have on the control of phytopathogenic microorganisms [42,43,44]. These natural compounds are commonly found in essential oils: eugenol and isoeugenol are found in *Syzygium aromaticum* and *Cinnamonum zeylanicum* [42,45,46] and thymol and carvacrol are found in *Thymus vulgaris* L. (Lamiaceae) and *Origanium vulgaris* [16,40,47]. They are considered safe substances by the European Commission and the U.S. Food and Drug Administration (FDA) and have even been registered as food seasoning [48,49].

A large selection of essential oils from medicinal plants and their majority compounds was previously published by Rozwalka et al. 2020 [36] showing effect to control disease of grapes. Therefore, this study aimed to assess the antifungal activity of eugenol, isoeugenol, thymol, carvacrol, and some of their ester derivatives in search of alternatives to inhibit *Colletotrichum nymphaeae* and *Colletotrichum chrysophillum* and to control GLS and BR, caused by two prevalent species in the main production apple areas of Brazil.

## 2. Experimental Part

### 2.1. Preparation of the Ester Derivatives

Compounds were synthesized using commercial reagents obtained from Sigma-Aldrich^®^, St. Louis, MO, USA. These reagents were used as received or purified when necessary. All solvents were subjected to fractional distillation prior to use. When required, solvents were dried using standard procedures. All reactions were monitored by thin-layer chromatography (TLC, on silica gel GF_254_) through comparison with the patterns of the starting substances. Figure 1 shows all structures of the commercially purchased substances and synthesized compounds.

Flash chromatography [50] or Dry-Column Flash Chromatography [51] was employed to purify the synthesized compounds (silica gel, 200–300 mesh). The reactions were carried out in anhydrous medium under argon atmosphere.

### 2.2. Characterization of the Ester Derivatives

The synthesized substances were characterized by gas chromatography coupled with mass spectrometry (GCMS) (QP2010 Plus, SHIMADZU, Kyoto, Japan) and hydrogen (^1^H) and carbon (^13^C) nuclear magnetic resonance (NMR) (4.7 tesla, Bruker, Rheinstetten, Germany), at room temperature using tetramethylsilane as an internal standard, with CDCl_3_ as solvent.

### 2.3. General Procedure for Acylation of the Phenolic Compounds

Based on the methodology described by Morais et al. (2014) [52], the phenolic compound (26 mmol) and pyridine (39 mmol, 3.08 g, 3.14 mL) were added to a 100 mL flask under argon atmosphere. After 5 min of magnetic stirring, the corresponding anhydride (39 mmol of acetic, butyric or benzoic anhydride) was added as the acylating agent according to the ester derivative of interest. The mixture was left stirring at room temperature for 24 h. Ethyl ether (60 mL) was added to the reaction mixture, which was transferred to a separating funnel. The organic phase was washed successively with water (4 × 15 mL), H_2_SO_4_ 1 M solution (2 × 15 mL), NaOH 2 M solution (2 × 15 mL), and finally again with water (4 × 15 mL). The organic phase was dried with Na_2_SO_4_, and the solvent was evaporated under reduced pressure. The substances were purified by flash chromatography using a mixture of hexane: ethyl acetate (9:1) solvents. The yields, spectral data, and ^1^H NMR spectra obtained for each compound are available in the Appendix A.

### 2.4. General Procedure for Preparation of the Thymol and Carvacryl Butyrate

Based on the methodology described by Martin and Demerseman (1989) [53], Thymol or carvacrol (23 mmol) and pyridine (57.5 mmol, 4.54 g, 4.63 mL) were added to a 100 mL flask coupled to a reflux condenser under argon atmosphere. After 5 min of stirring, butyric anhydride (34.5 mmol, 5.45 g, 5.24 mL) was added to the reaction medium and the mixture was refluxed (120 °C) for 4 h. Subsequently, the mixture was allowed to cool to approximately 80 °C, distilled water (6 mL) was added, the temperature was again raised to 120 °C, and the mixture was refluxed for another 30 min. The system was allowed to cool, and the products were extracted as previously described in the general procedure for acylation of the phenolic compounds. The organic phase was dried with Na_2_SO_4_, and the solvent was evaporated under reduced pressure. There was no need for further purification of the products. The yields, spectral data, and ^1^H NMR spectra obtained for each compound are available in the Appendix A.

### 2.5. Mycelial Growth Assay

The fungal isolates *Colletotrichum nymphaeae* (isolate Ca32) and *Colletotrichum chrysophillum* (isolate Col33) from the LEMID-UFPR collection, originally isolated from GLS lesions on leaves, were transferred to potato dextrose agar (PDA) medium. The cultures were incubated at 23 °C under a 12-h photoperiod for 96 h to obtain young and actively growing colonies.

Incorporation of fungicides and the products to be tested in the fluxing medium was used for the mycelial growth assay—a technique widely employed in studies in the field of phytopathology [10]. In this technique, the compounds tested were previously prepared in stock solutions of known concentration. Dimethyl sulfoxide (DMSO) was used as solvent, and Adyvex OP 110^®^ (SGS Polímeros, São Sebastião do Caí, Brazil) surfactant was utilized to assist with the subsequent suspension of the compound in the culture medium. An aliquot of the stock solution was then added to 40 mL of sterile liquid PDA medium to obtain the final concentration of interest. The PDA medium carrying the compound in the previously stipulated concentration (25, 50, 75, 100, 125, 200, 300, 600, 800, 1000, 1200, or 1400 mg L^−1^) was poured into Petri dishes and left to solidify at room temperature. Sequentially, a 5 mm mycelial plug, from both *Colletotrichum* species, was removed from the colony border after 96 h of incubation and deposited on the PDA medium containing the compound to be tested. Dishes containing only PDA and containing PDA plus solvent and surfactant (1.5 mL and 7 μL, respectively) were used as controls. Dishes containing the fungicide Thiophanate-methyl commercially obtained (Cercobin^®^ 700 wp, IHARABRAS S/A Indústrias Químicas, Sorocaba, Brazil) at the concentration of 1 mg·L^−1^ were also used as controls.

The Petri dishes were incubated at 23.5 °C at a 12 h photoperiod. Colony diameters were measured when the mycelial mass in the control Petri dishes, with only PDA, reached 80% of the plate diameter (approximately after 72 h of incubation). All compounds and their respective concentrations were tested in quadruplicate. The percent inhibition of mycelial growth caused by the treatment was calculated using the following formula:$$\%\;inhibition=\left(\frac{d-D}{d}\right)\cdot 100$$
where: *d* = colony diameter in control; *D* = colony diameter in treatment.

Homogeneity of variance was verified by the Bartlett test, then the data were submitted to analysis of variance (ANOVA), verifying the effect of treatments by the F test for all assays. When statistically significant difference was observed, the means were compared using the Scott-Knott test (*p* < 0.05). There were no transformation data for statistical analysis in any of these experiments.

### 2.6. Minimum Inhibitory Concentration (MIC)

The isolates (Ca32 and Col33) were transferred to culture medium containing oats (10%) and PDA (1.5%) and incubated at 23 °C at a 12 h photoperiod, for 168 h, to obtain colonies in the sporulation stage.

The MIC assay was conducted by the broth microdilution method according to the National Committee for Clinical Laboratory Standards [54] protocol with modifications. The compounds tested were previously prepared in stock solutions of known concentration using DMSO as solvent.

To serve as negative control, 160 μL of PDA culture medium were added to a 96-well microdilution plate. As positive control, 160 μL of the same culture medium containing spores of the tested fungi (inoculum) at the concentration of 10^5^ conidia/mL were added.

The compounds to be tested as antifungal agents were added to the microdilution plates in a gradient of six concentrations in triplicate. The concentration gradient was obtained by adding an aliquot of the stock solution of each compound aiming to obtain the final test concentrations (25, 50, 75, 100, 125, 150, 175, 200, 250, 300, 400, 425, 550, 600, 675, 800, 1000, 1200, or 1400 mg L^−1^). Wells containing only DMSO were prepared with the amounts used during the test (2.5%. 5%. 7.5%. 10%. 12.5%, and 15% *v*/*v*). Culture medium (160 μL PDA), with inoculum, were also added to all the wells containing the compounds. Sterile water was added to all the plate wells to complete the final volume of 200 μL.

Commercial fungicide Manzate-800^®^ (UPL do Brasil, Ituverava, Brazil) was also used as a control at concentrations of 0.0016, 0.08, 0.4, 2, 10, and 50 mg L^−1^ of the active ingredient (mancozeb) in triplicate.

The microdilution plates were incubated at 23 °C at a 12 h photoperiod. Optical density (OD) of the microdilution plates, was using a microdilution plate spectrophotometer (Bio Tek, model Elx 800 GIDX, 5–96 wells, Winooski, VT, USA) equipped with a 405 nm filter. Readings were performed at 24 and 168 h of incubation.

The MIC was assessed by comparing the OD of the wells containing the tested compounds with those of the controls. Wells with OD like that presented by the negative control indicated no biological activity, whereas wells with OD greater than that presented by the negative control indicated biological activity. The MIC was obtained considering the value found in the concentration prior to the first well of the plate in which fungal viability was observed.

The previously described procedure was used in two MIC assays, performed independently, named Exp. 1 and Exp. 2.

### 2.7. Ex Vivo Assay

The most promising compounds from the in vitro tests were used for these assays. Thus, two ex vivo tests were carried out to evaluate the fungicidal effect of these compounds in the control of GLS (on leaves) and ABR (on fruit). The minimum concentration at which the substance was able to completely inhibit the mycelial growth and/or germination of conidia, of both *Colletotrichum* species, were the concentrations adopted for each treatment in the ex vivo assays.

The isolates (Ca32 and Col33) were transferred to a culture medium containing PDA and incubated at 25 °C with a 12-h photoperiod for 7 days, and then used for leaf and fruit inoculation.

The compounds slated for evaluation were weighed and dissolved in 1 mL of DMSO. To facilitate subsequent suspension of the compounds in an aqueous medium, the surfactant Adyvex OP 110^®^ (7% *w*/*w*) was employed. Eppendorf tubes containing the pre-prepared compounds were then transferred to a vessel containing 350 mL of water, yielding the final treatment concentrations. Untreated leaves and fruits were utilized as controls. Additionally, fruit samples were treated with a solution comprising 0.35% DMSO and Adyvex OP 110^®^ as an additional control. For the treatment conditions, two dosages of Manzate-800^®^ (100 and 200 mg L^−1^) were also applied to treat both leaves and fruits.

Apple fruits of the Gala cultivar were employed, undergoing a superficial disinfection process involving 70% ethanol and 0.5% sodium hypochlorite treatment for 1 min, followed by triple rinsing with sterile distilled water. Subsequently, using a needle, each fruit was incised to a mean depth of 0.5 cm. The fruits were individually immersed in treatment suspensions for 30 s, followed by air-drying at room temperature. Two hours post-treatment, each fruit was inoculated with a 5 mm diameter mycelial disc and placed in sterile, plastic containers, which were moistened to maintain humidity. Incubation took place at 25 °C under a 12-h photoperiod.

The experiment was duplicated, employing a completely randomized design with 12 fruits per treatment and isolate. Daily assessments were conducted on the fruits to monitor the onset of initial symptoms and signs. After a 10-day incubation period, lesion diameter measurements were taken.

Leaves of the Gala cultivar were collected from field-grown plants. The leaves were immersed in a container of distilled water for 1 min to eliminate any potential surface residues. Subsequently, they were subjected to immersion treatment, dipped into a suspension of each treatment for 30 s, and air-dried at ambient temperature. Spore suspensions were prepared for the isolates *C. nymphaeae* and *C. chrysophilum*, adjusted to a concentration of 1 × 10^4^ conidia mL^−1^.

Two hours post-treatment, the leaves were inoculated with mixed suspensions of the isolates, sprayed on both the abaxial and adaxial surfaces. The leaves were then placed in Gerbox^®^ (J.Prolab, São Paulo, Brazil) containers with sterilized water at the bottom to maintain humidity. A mesh was used to prevent direct contact with the water. The conversion 3.3.2 tainers were placed in a BOD incubator at a temperature of 25 °C under a 12-h photoperiod.

The experiment was conducted in duplicate using a completely randomized design, featuring 4 replicates per treatment. Each replicate consisted of 3 leaves. Daily observations were made on the leaves to monitor the appearance of initial symptoms and signs. After 13 days of incubation, disease severity was assessed using a diagrammatic scale proposed for the specific pathosystem [55].

Using symptom and sign data, the incubation period and latency period were determined for GLS and ABR on leaves/fruits. A survival analysis was conducted, considering the probability of leaves/fruits remaining asymptomatic or without producing signs over time. Kaplan-Meier curves were constructed to assess the probability of symptom appearance, and a semi-parametric Cox model was fitted to compare the curves. The analyses were performed using the R software version 3.3.2 (R Development Core Team 2016), the package ‘survival’ was used for survival analyses.

The incidence of symptomatic leaves was estimated for each treatment by the ratio between the number of leaves with GLS symptoms and the total leaves assessed. The GLS severity on leaves was estimated by calculating the average percentage of diseased area considering all leaves of each treatment with a standard area diagram set developed for this disease [55]. For fruit essays the incidence (number of diseased fruits in total fruit assessed) was quantified every day for 10 days. Also, the average of ABR lesion diameter was measure at 10 days after the inoculation (end of the experiment)

Homogeneity of variance was verified by the Bartlett test, then each data was submitted to analysis of variance (ANOVA), verifying the effect of treatments by the F test for the two assays. When statistically significant difference was observed between the treatments, the means were compared using the Tukey’s test (*p* < 0.05). The analyses were performed using the R software, version 3.3.2 (R Development Core Team 2016).

## 3. Results

### 3.1. Synthesis of the Esters Derived from Phenolic Substances

All esters were synthesized using classical methodologies and obtained in yields that varied from good, in the case of the esters Eugenyl acetate (EgAc), Isoeugenyl acetate (IegAc), Thymol acetate (TmAc), Thymol butyrate (TmBt), Carvacryl acetate (CvAc) and Carvacryl butyrate (CvBt) to reasonable, in the case of the esters Eugenyl butyrate (EgBt), Eugenyl benzoate (EgBz), Isoeugenyl butyrate (IegBt), Isoeugenyl benzoate (IegBz), Thymol benzoate (TmBz), and Carvacryl benzoate (CvBz). After purification, the target substances were obtained with a high level of purity.

### 3.2. Mycelial Growth Assay

Table 1 shows the average of the results obtained for the mycelial growth test performed with the precursor compounds and their derivatives against the *C. nymphaeae* and *C. chrysophillum* fungi. The DMSO (solvent)/sulfactant treatment added to the culture medium showed inhibition of 33.8 ± 11.9% for *C. nymphaeae* and 28 ± 5.8% for *C. chrysophillum*. This treatment was not used to compare treatments, just as a reference.

Mycelial growth inhibition capacity of eugenol (Eug) and its derivatives showed a well-defined pattern, and it could be clearly observed that effectiveness decreased as the size of the ester portion in the structure of the substituent groups (EgAc, EgBt, and EgBz) increased. Comparison between the inhibition capacity of Eug and EgAc showed that they present similar activities (approximately 84% inhibition at a concentration of 300 mg L^−1^), demonstrating that this structural modification, for this stage of fungus development, did not modify its activity.

The same trend observed for Eug and its derivatives was verified for isoeugenol (Ieg) and its derivatives (IegAc, IegBt and IegBz). It is also worth noting that the double bond conjugated to the aromatic ring in the isoeugenol structure (Ieg) attenuates its activity against the tested fungi.

In the present study, the results obtained for thymol (Tm), carvacrol (Cv), and their derivatives (TmAc, TmBt, TmBz, CvAc, CvBt, and CvBz) did not follow the same trend presented by Eug, Ieg, and their derivatives. TmBt presented activity like that of Tm. TmAc showed total mycelial growth inhibition at the concentration of 200 mg L^−1^, lower than that of thymol.

The structural changes in the Cv molecule decreased its activity, although the activities showed the same trend as those observed for the Tm derivatives (TmAc, TmBt and TmBz). In general, Cv and its derivatives (CvAc, CvBt and CvBz) showed smaller activity against the tested fungi compared with that of Tm and its derivatives.

The compounds used in this study have the advantage of not distinguishing between the two *Colletotrichum* species, considering that the same inhibitory activity was noticed for both species. This activity was not observed for the fungicide thiophanate-methyl, one of the main active ingredients recommended for the GLS control, which proved to be inefficient to control *C. nynphaeae* (−7.3 ± 9.2%), even though it was efficient to inhibit mycelial growth of *C. chrysophilum.*

### 3.3. Minimum Inhibitory Concentration (MIC)

The MIC values obtained for the investigated natural compounds and their ester derivatives using conidia of two species of the pathogen. The inhibitory activity against the conidia of both *Colletotrichum* species presented by Tm, Cv and its derivatives (TmAc, TmBt, TmBz, CvAc, CvBt and CvBz) kept the trend observed in the mycelial growth stage. Derivatization of the phenolic compounds to the acetate and butyrate esters did not cause a marked loss in their activities. TmBt showed the same MIC value as Tm and Cv against both *Colletotrichum* species (100 mg L^−1^); CvBt, CvAc and TmAc were a little less effective (200 mg L^−1^). However, modification of the phenolic hydroxyl to benzoate ester caused significant loss in the activities of the CvBz and TmBz, presenting MIC only with 1000 mg L^−1^.

Eg and EgAc proved to be more active at a specific stage of fungal development. At the concentration of 300 mg L^−1^, both compounds completely inhibited mycelial growth. In the conidia germination, even with the highest concentration tested (675 mg L^−1^), germination inhibition was not observed in the tested fungi.

It is worth mentioning that Ieg was active against conidia germination (Table 2). The lowest concentration tested (400 mg L^−1^) totally inhibited the activity of both *Colletotrichum* species. The other Eg and Ieg derivatives (EgBt, EgBz, IegAc, IegBt and IegBz) were much less active against the fungi tested.

## 4. Ex Vivo Assays

The estimates of relative risk for the expression of symptoms of apple bitter rot caused by *C. nymphaeae* (Appendix A) or *C. chrysoplhilum* (Appendix A) are highlighted for treatments statistically differed from the control. Fruit treated with carvacryl acetate (CvAc), eugenyl acetate (EgAc), carvacryl butirate (CvBt), Eugenol (Eug), Manzate-800 at a concentration, 100 and 200 mg L^−1^ (Mzeb100 and Mzeb 200, respectively), somehow were the ones that presented the lowest relative risk of presenting infection by *C. nymphaeae*, so, they have the lowest probability of the apple fruit show symptoms. Nevertheless, in the repetition of the experiments only thymol (Tm) confirms the results of the first experiment. On second experiment, eugenol (Eug) treatment had the low and significant relative risk of presenting infection.

The treatments that significantly delayed the onset of symptoms on fruit caused by *C. chrysophillum* were carvacryl acetate (CvAc), eugenyl acetate (EgAc), carvacrol (Cv), carvacryl butirate (CvBt), isoeugenol (Ieg), Manzate-800, 100 mg L^−1^ (Mzeb 100) and thymol butyrate (TmBt). Isoeugenol treatment repeat its effect in the repetition ex vivo assay and eugenol (Eg) appear (Appendix A).

It is important to point out that the Manzate-800 treatments (Appendix A), which is recommended for the control of ABR, were efficient against *C. nymphaeae* only at the first experiment. Its effect was not repeated in the second experiment. Then, when compared to control treatment, at concentrations 100 and 200 mg L^−1^, Manzate-800 failed BR control when the inoculation was done with *C. chrysophilum* in both experiments (Appendix A).

The lesion diameter caused by *C. nymphaeae* (Table 3) were significantly lower in Eg, EgAc and CvAc compared to the control. In addition, Cv, Mzeb 100, Mzeb 200 and TmBt, also showed significantly lower values compared to control treatment. The lesion diameter after inoculation with *C. chrysophillum* (Table 3) were significantly lower in CvAc, TmBt, Eg and Ieg. The conventional fungicides treatments (Mzeb 100 and Mzeb 200) showed high lesion diameter and failed to control ABR. DMSO did not differ from control, and, in some cases, it also does not differ from treatments.

In the second kind of ex vivo assay (with mixture of *Colletotrichum nymphaeae* and *Colletotrichum chrysophillum*), the percentage of leaves with symptoms of GLS in the untreated control was 91.6%. All product tested showed significant lower incidence comparing to control with values varying from zero to 41.66% of diseased fruit. The severity, area of GLS symptoms in the leaves, was 34.17% and 70% in the untreated control leaves while in the treatments the severity varied from zero (several treatments) to 15.32% (Tm) (Table 4).

For the same essay it was estimates of relative risk for the expression of symptoms of Glomerella Leaf Spot occurs on leaves with the mixture of isolates assessing the symptoms over the time (Appendix A). All treatments exhibited a significantly low risk of presenting GLS symptoms. The incubation period (number of days for symptoms expression) or latent period (number of days for sporulation of the pathogen) were 4–6 and 12 days, respectively, in the control treatment. For most of products tested, few symptoms were observed and the time for symptoms and sporulation appearence increased to over then 13 days (Appendix A). The Figure 2 represents the summary of the key points addressed in this research.

## 5. Discussion

Thymol, carvacrol, and thymol butyrate were equally effective in controlling the two species of *Colletotrichum* in both infection and colonization stages by conidium germination and mycelium growth interferences, respectively. In the conidial germination stage, the results of some treatments were comparable to those of commercial fungicide Manzate-800^®^. Symptoms of bitter rot on fruit were reduced by eugenol (Eg), eugenyl acetate (EgAc), carvacrol (Cv), carvacryl acetate (CvAc), isoeugenol (Ieg) and thymol butyrate (TmBt), which significantly delayed the onset of the disease. For GLS on lea, all treatments showed a good effect in reducing the disease on detached leaves experiments.

The inhibition of mycelial grow by eugenol against both *Colletotrichum* species is consistent with previous findings that have similarly documented the fungicidal effect of eugenol on the mycelial growth of phytopathogenic fungi. Yang et al. (2020) [56] found EC_50_ values of 190.58 mg L^−1^ and 42.04 mg L^−1^ of eugenol against *Fusarium graminearum* and *Valsa mali*, respectively. Lima et al. (2022) [57] reported the total mycelial growth inhibition of *Colletotrichum sp.*, isolated from a papaya fruit with typical symptoms of anthracnose, with >500.00 mg L^−1^ of eugenol. Eugenol also presented excellent vapor phase activity, found by Quintana-Rodriguez et al. (2018) [58], that showed the total inhibition of the *Colletotrichum lindemuthianum, Fusarium oxysporum* and *Botrytis cinerea* mycelial growth with 0.01 mg L^−1^ of eugenol.

Fungicidal efficacy of isoeugenol against the mycelial growth of *Colletotrichum graminicola* and *Fusarium solani* was also reported by Dev et al. (2004) [59]. The same authors compare eugenol with isoeugenol. Eugenol, at the concentrations of 270 and 340 mg L^−1^, completely inhibited the mycelial growth of *C. graminicola* and *F. solani*, respectively; while isoeugenol inhibited the mycelial growth of *C. graminicola* and *F. solani* only at the concentrations of 4760 and 4920 mg L^−1^, respectively. The result corroborates our findings.

The activity presented for thymol and carvacrol, against both *Colletotrichum* species to inhibit mycelial grow also agree with previous findings reported in the literature. Thymol and carvacrol demonstrated complete inhibition of the mycelial growth of *C. acutatum* at a concentration of 150 mg·L^−1^ [60]. Under the same conditions both, thymol and carvacrol, totally inhibit the mycelial growth of *Botryodiplodia theobromae* at the concentration of 150 mg·L^−1^ [61]. So, ester derivatives of thymol and carvacrol could be as active as their precursors. The researchers reported that Carvacryl acetate and thymol acetate inhibit >96% of mycelial growth with 50 mg L^−1^ against two plant pathogenic fungus (*Botrytis cinerea* and *Rhizoctonia solani)*. At the same concentration, carvacrol and thymol reached between 80 to 90% for the same pathogens.

Regarding the inhibition of conidial germination (MIC assay), the findings elucidated by this study corroborates Scariot et al. (2020) [62], who investigated the efficacy of twenty monoterpenoids against *C. fructicola* and *C. acutatum* to ascertain their effectiveness. The results showed that thymol and carvacrol were the most effective among them. On *C. acutatum*, thymol and carvacrol at 125 mg L^−1^, were sufficient to inhibit ≥90% conidial germination. Also, Carvacrol at 125 µL L^−1^ completely inhibit the conidial germination of *C. fructicola* for 18 h [63].

In agreement to our results, low activity of eugenol against *Colletotricum musae* conidial germination was reported previously [64]. The concentrations of eugenol needed to totally inhibit the germination of conidia of *C. musae* and *Fusarium proliferatum* were 0.14% (1400 μL L^−1^) and 0.12% (1200 μL L^−1^), respectively.

The strong antifungal and antibacterial activity shown by monoterpenoids is mainly related to their ability to disrupt the integrity of the cell membrane [62,63,65,66]. As verified in the results herein exposed, Tm, Cv, Eg and Ieg presented fungicidal activity equal to or greater than those of their ester derivatives. Many researchers have attributed the strong antibacterial and antifungal activity of natural compounds such as thymol, carvacrol, and eugenol to the presence of phenolic hydroxyl [60,66,67,68,69].

The antimicrobial activity of essential oils has a positive correlation with the number of phenolic substances present in these oils [68]. Essential oils with high content of phenolic compounds present greater antifungal activity than other organic functions, such as non-aromatic alcohols, aldehydes, and hydrocarbons. Ultee et al. (2002) [66] attributed the mode of action and the strong activity of these compounds to the phenolic hydroxyl, which acts as a proton exchanger, thus reducing the pH gradient across the cell membrane. Many authors agree with this observation, such as Veldhuizen et al. (2006) [69], who tested compounds with structure like that of carvacrol and found that removal of the phenolic hydroxyl from the structure of carvacrol caused a significant loss in its bactericidal activity, although the researchers noticed that the substitution of the phenolic hydroxyl by an amine group, an equally polar organic function, does not cause much damage to the activity of carvacrol-like compound. The process of detoxification, presented by *C. acutatum* and *B. theobromae* [60] also demonstrate that methylation and acetylation of the hydroxyl group in the aromatic ring, added aromatic and aliphatic hydroxylation, represent the main metabolic pathway to make thymol and carvacrol less toxic.

Nevertheless, studies have also shown that derivatization of the phenolic hydroxyl for the ester and ether groups can increase the activity of compounds such as those investigated in this study, attributing a less important role to phenolic hydroxyl. Wang et al. (2019) [61], observed that the acetylation and methylation of the phenolic hydroxyl increased the fungicidal activity of thymol and carvacrol, they also attributed their activity not only to the phenolic hydroxyl, but also to the aliphatic groups present in its structure, which confers an adequate hydrophobicity for them. Xie et al. (2017a) [38] reported that methylisoeugenol has greater fungicidal activity than isoeugenol, and that methyleneugenol is more active than eugenol against *R. solani*. In the control of *F. oxysporum*, methylisoeugenol and isoeugenyl acetate were more active than isoeugenol.

The structure/activity relationships presented by the compounds analyzed in this study corroborate the conclusions reached by Carrasco et al. (2012) [70], who used 21 natural and synthetic phenylpropanoids with structures analogous to that of eugenol (including safrole and isoeugenol) to assess their antifungal properties against a variety of human opportunistic pathogenic fungi. Comparison of compounds with similar structures showed that the presence of the allyl group, present in eugenol, absent in isoeugenol (Figure 2) is very important for the antifungal characteristic of these phenylpropanoids analogous to eugenol. The same research concluded that the phenolic hydroxyl had no influence on the antifungal activity of the eugenol analogs, considering that the hydroxyl esterified in the form of acetate did not cause damage to the final activity of the compared compounds.

The best inhibition results, verified by the MIC values for the tested compounds, shown in Table 2, are relatively close to the inhibition concentration of mancozeb, showing that the results found in these experimental conditions are promising. This type of compounds, such as natural products and their ester derivatives, can be a suitable alternative for the management of phytopathogens and at the same time, generate less damage to the environment.

Investigations regarding the effectiveness of derivatives from natural products, such as ester derivatives, against phytopathogenic fungi, in ex vivo assay, is extremely scarce. The natural products eugenol, thymol and carvacrol have been evaluated and present positive results in ex vivo assays and in field conditions. Jing et al. (2017) [71] reported that eugenol showed in field condition, high efficiency control of black shank, disease that affect tobacco culture, which is caused by the fungus *Phytophthora nicotianae*. Demonstrating results remarkably like those of the positive control, metalaxyl-Mn-Zn, a systemic fungicide. In an ex vivo assay, Zhou et al. (2017) [44] tested eugenol in the vapor phase, during post-harvest storage, against the development of soft root disease in peaches inoculated with the fungus *Rhizopus stolonifer*, in which it was able to significantly reduce the lesions caused by the disease, as well as, had a low incidence of it.

Thymol also has positively effect on phytopathogen control. Chillet et al. (2019) [72] prepare a thymol solution with a penetration agent in the fruit to control anthracnose in mango, caused by *C. gloeosporioides*. The thymol best concentration treatment was 0.1% (1000 mg L^−1^), and it was capable to totally controlled both wound anthracnose and natural quiescent anthracnose during the 7-day storage period. At the same time, this thymol treatment solution concentration had no detectable effect on fruit maturation and quality.

Carvacrol also shows disease control, Martínez-Romero et al. (2007) [73], used carvacrol in vapor phase to control *B. cinerea* inoculated in table grape in stored packages. Carvacrol presented dose-dependent effect, controlling fungal decay with 81 ± 3 and 93 ± 1% effectiveness for 0.5- and 1.0-mL L^−1^ and had little effect on the physiological parameters of table grapes, compared to the control.

The results of the in vitro and ex vivo assays differed in terms of the activity shown by the treatments with the natural compounds. This fact may be related to the large increase in variables found in the ex vivo assay. For example, the possibility of the treatments presenting phytotoxicity, the fungi being more adapted to use the leaf or fruit material as a substrate than the culture medium, also the loss of the active principle by volatilization, since the humid chamber does not remain hermetically sealed. In a way, these same limitations can interfere with the reproducibility between ex vivo assays, explaining some differences. Using active principles nanoencapsulation technology could be one option to improve efficiency by natural compounds for future studies.

The most used fungicide for the control of GLS and BR, contains the active ingredient mancozeb, which was classified, by the Risk Assessment Committee (RAC) of the European Chemical Agency (ECHA), as a substance “toxic for reproduction 1B” (R1B), for presenting severe brain malformations caused by ethylene thiourea, one of its metabolites [74]. Starting from this classification, the European Commission published on 14 December 2020, the Implementing Regulation 2020/2087 regarding the non-renewal of the active substance mancozeb in the Official Journal [75]. So, although the natural product and ester derivatives presented some limitations under the experimental conditions, proposing new alternatives to mancozeb is increasingly important. The natural compounds used in this study are considered safe substances by the European Commission and the U.S. Food and Drug Administration (FDA) and can be used as food seasoning.

A possible limitation of the use of the components testes is the sensitivity of esters to hydrolysis under extreme pH conditions, potentially compromising the fungicidal activity of the esterified derivatives. However, throughout the study, this did not occur. For future field studies, it is recommended to monitor pH levels to ensure product efficacy, especially if used in mixtures. Additionally, even if partial hydrolysis of the esters were to occur, the phenolic components resulting from hydrolysis exhibit strong antifungal activity. For the continuation of this research, studies of mixed formulations between the components with effect on the germination of conidia and mycelial growth are recommended. Based on the formulation generated, field studies must be carried out since flowering, when the pathogen infects the flower and can extend to the fruits, as well as applications at harvest. These two phases are critical for the sustainable management of BR in fruits and GLS in leaves and would minimize the effects on beneficial organisms such as bees and protect the consumer from the excessive use of fungicides on fruit. Another perspective for using these products would be as a sanitization option to reduce primary inoculum at the end of winter and the beginning of bud break.

## Figures and Tables

**Figure 1 plants-13-03196-f001:**
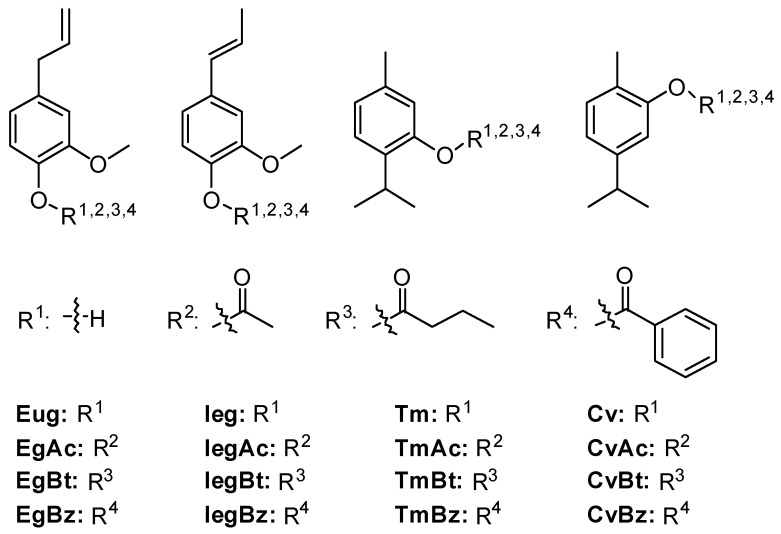
Natural compounds and derivatives used in this study.

**Figure 2 plants-13-03196-f002:**
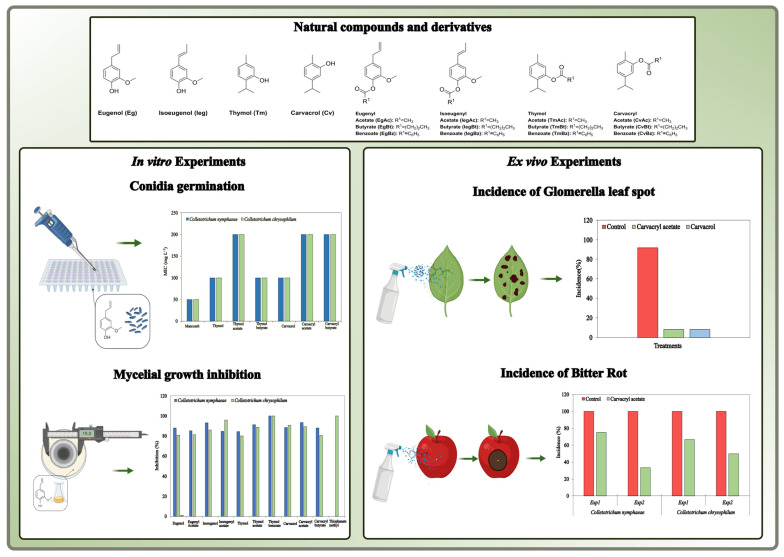
Summary of the key points addressed in this work.

**Table 1 plants-13-03196-t001:** Percentage of mycelial growth inhibition of *C. nymphaeae* and *C. chrysophillum* in different concentrations of natural products and their derivatives.

Treatments *	mg L^−1^	Inhibition (%) **				Treatments *	mg L^−1^	Inhibition (%) **			
		*C. nymphaeae*		*C. chrysophillum*				*C. nymphaeae*		*C. chrysophillum*	
Thiophanate-Methyl	1	−7.3 ± 9.2		100 ± 0.0	a	DMSO/ Surfactant	0.04	33.8 ± 11.9		28 ± 5.8	
Eg	600	100.0 ± 0.0	a **	100.0 ± 0.0	a	Tm	125	100.0 ± 0.0	a	100.0 ± 0.0	a
Eg	300	87.9 ± 5.2	b	80.8 ± 7.3	c	Tm	50	84.4 ± 1.4	b	80.1 ± 0.8	c
Eg	200	73.0 ± 11.0	c	77.8 ± 4.0	c	Tm	20	67.0 ± 15.7	c	57.1 ± 17.7	e
EgAc	600	100.0 ± 0.0	a	100.0 ± 0.0	a	TmAc	200	100.0 ± 0.0	a	100.0 ± 0.0	a
EgAc	300	85.2 ± 6.6	b	84.1 ± 4.9	b	TmAc	125	91.3 ± 4.2	b	88.6 ± 5.6	b
EgAc	200	72.4 ± 4.2	c	62.5 ± 3.5	d	TmAc	50	45.9 ± 3.8	e	43.6 ± 8.5	f
EgBt	1000	76.4 ± 1.1	c	70.4 ± 2.6	d	TmBt	200	100.0 ± 0.0	a	100.0 ± 0.0	a
EgBt	800	72.5 ± 2.8	c	56.0 ± 6.2	e	TmBt	125	100.0 ± 0.0	a	100.0 ± 0.0	a
EgBt	600	70.7 ± 2.2	c	64.4 ± 3.2	d	TmBt	50	62.3 ± 5.4	d	45.8 ± 16.8	f
EgBz	1200	44.5 ± 1.1	e	45.6 ± 2.8	f	TmBz	1200	57.7 ± 3.8	d	41.2 ± 3.8	f
EgBz	1000	50.0 ± 2.9	d	57.4 ± 3.0	e	TmBz	1000	73.4 ± 1.5	c	66.3 ± 2.5	d
EgBz	600	33.9 ± 7.4	f	51.9 ± 5.5	e	TmBz	600	72.6 ± 3.4	c	63.6 ± 2.5	d
Ieg	1000	88.5 ± 13.3	b	82.4 ± 4.4	c	Cv	125	100.0 ± 0.0	a	100.0 ± 0.0	a
Ieg	800	90.1 ± 19.8	b	68.1 ± 27.8	d	Cv	60	88.5 ± 11.0	b	90.7 ± 11.1	b
Ieg	600	93.1 ±13.8	b	86.1 ± 9.6	b	Cv	50	70.6 ± 3.2	c	69.1 ± 2.9	d
IegAc	1000	90.8 ± 6.5	b	86.6 ± 1.8	b	CvAc	600	100.0 ± 0.0	a	100.0 ± 0.0	a
IegAc	800	95.6 ± 8.8	b	86.8 ± 7.2	b	CvAc	200	93.5 ± 7.8	b	89.4 ± 8.3	b
IegAc	600	84.5 ± 12.4	b	95.8 ± 8.3	a	CvAc	125	79.7 ± 11.3	c	76.3 ± 11.0	c
IegBt	1400	56.6 ± 10.5	d	64.3 ± 4.9	d	CvBt	200	100.0 ± 0.0	a	100.0 ± 0.0	a
IegBt	1200	65.9 ± 4.2	d	54.9 ± 9.6	e	CvBt	125	87.9 ± 2.0	b	80.5 ± 1.7	c
IegBt	1000	56.9 ± 6.6	d	66.2 ± 5.5	d	CvBt	50	57.1 ± 1.7	d	49.6 ± 0.8	f
IegBz	1000	39.1 ± 9.5	f	48.1 ± 3.7	f	CvBz	1000	48.6 ± 5.3	e	62.3 ± 2.5	d
IegBz	600	21.3 ± 6.0	g	36.6 ± 4.1	g	CvBz	600	36.7 ± 5.5	f	46.5 ± 1.3	f
IegBz	200	17.2 ± 7.7	g	25.0 ± 1.9	g	CvBz	200	29.7 ± 6.0	f	30.6 ± 3.9	g

* Eugenol (Eug); Eugenyl acetate (EgAc); Eugenyl butyrate (EgBt); Eugenyl benzoate (EgBz); Isoeugenol (Ieg); Isoeugenyl acetate (IegAc); Isoeugenyl butyrate (IegBt); Isoeugenyl benzoate (IegBz); Thymol (Tm); Thymol acetate (TmAc); Thymol butyrate (TmBt); Thymol benzoate (TmBz); Carvacrol (Cv); Carvacryl acetate (CvAc); Carvacryl butyrate (CvBt); Carvacryl benzoate (CvBz). ** Original data, without processing; means followed by different letters in the column differ significantly by the Scott-Knott test at 5% probability.

**Table 2 plants-13-03196-t002:** Minimal inhibitory concentration (MIC) of natural products and their derivatives against *C. nymphaeae* and *C. chrysophillum.*

Treatments *	MIC *C. nymphaeae*		MIC *C. chrysophillum*	
	Exp. 1 **	Exp. 2 **	Exp. 1 **	Exp. 2 **
	mg L^−1^	mg L^−1^	mg L^−1^	mg L^−1^
Mancozeb	50	50	>50	50
DMSO	100,000	100,000	100,000	100,000
Eg	>675	>675	>675	>675
EgAc	>675	>675	>675	>675
EgBt	>1400	>1400	>1400	1000
EgBz	1000	1000	1000	1000
Ieg	<400	600	600	<400
IegAc	1200	1200	1200	1200
IegBt	600	>1400	>1400	800
IegBz	1000	1000	>1400	>1400
Tm	100	100	100	100
TmAc	200	200	200	200
TmBt	100	100	100	100
TmBz	1000	1000	1000	1000
Cv	100	100	100	100
CvAc	200	200	200	200
CvBt	200	200	200	200
CvBz	1000	1000	>1400	1000

* Eugenol (Eg); Eugenyl acetate (EgAc); Eugenyl butyrate (EgBt); Eugenyl benzoate (EgBz); Isoeugenol (Ieg); Isoeugenyl acetate (IegAc); Isoeugenyl butyrate (IegBt); Isoeugenyl benzoate (IegBz); Thymol (Tm); Thymol acetate (TmAc); Thymol butyrate (TmBt); Thymol benzoate (TmBz); Carvacrol (Cv); Carvacryl acetate (CvAc); Carvacryl butyrate (CvBt); Carvacryl benzoate (CvBz). ****** Experiments 1 and 2 follow the same methodology and were conducted at separate time points with 30 days apart.

**Table 3 plants-13-03196-t003:** Lesion diameter and Incidence of bitter rot, caused for *Colletotrichum nymphaeae* or *Colletotrichum chrysophillum.*

*Colletotrichum nymphaeae*									
Experiment 1 ***					Experiment 2 ***				
Treatment *	Lesion Diameter (mm)		Incidence (%)		Treatment *	Lesion Diameter (mm)		Incidence (%)	
	10DAI		10DAI			10DAI		10DAI	
Control	110.33	a **	100.00	a	Control	66.83	b	100.00	a
CvAc	71.36	bc	75.00	b	CvAc	83.25	a	33.33	c
EgAc	55.63	c	50.00	b	EgAc	93.83	a	75.00	b
TmAc	76.79	abc	83.33	a	TmAc	83.50	a	75.00	b
CvBt	86.58	abc	83.33	a	CvBt	88.50	a	75.00	b
Cv	66.08	bc	91.66	a	Cv	92.42	a	50.00	c
DMSO	79.75	abc	100.00	a	DMSO	86.08	a	83.33	a
Eug	61.63	bc	83.33	a	Eug	37.33	c	75.00	b
Ieg	95.64	ab	91.60	a	Ieg	82.00	a	75.00	b
Mzeb 100	73.04	bc	100.00	a	Mzeb 100	83.25	a	66.66	b
Mzeb 200	71.00	bc	83.33	a	Mzeb 200	82.92	a	66.66	b
TmBt	58.18	c	83.33	a	TmBt	70.58	b	75.00	b
Tm	89.18	abc	91.66	a	Tm	70.00	b	75.00	b
** *Colletotrichum chrysophillum* **									
**Experiment 1 *****					**Experiment 2 *****				
**Treatment ***	**Lesion Diameter (mm)**		**Incidence (%)**		**Treatment ***	**Lesion Diameter (mm)**		**Incidence (%)**	
	**10DAI**		**10DAI**			**10DAI**		**10DAI**	
Control	97.70	ab	100.00	a	Control	71.04	b	100.00	a
CvAc	51.77	c	66.66	c	CvAc	71.54	b	50.00	c
EgAc	80.63	abc	83.33	b	EgAc	107.29	a	91.66	a
TmAc	78.96	abc	91.66	a	TmAc	89.50	ab	75.00	a
CvBt	78.54	abc	91.66	a	CvBt	78.46	ab	91.66	a
Cv	73.00	abc	83.33	b	Cv	85.33	ab	50.00	c
DMSO	64.00	bc	83.33	b	DMSO	66.42	b	66.66	b
Eug	101.64	a	91.66	a	Eg	56.00	c	58.33	c
Ieg	60.38	c	75.00	c	Ieg	70.79	b	83.33	a
Mzeb 100	80.75	abc	91.66	a	Mzeb 100	85.75	ab	41.66	c
Mzeb 200	81.29	abc	91.66	a	Mzeb 200	79.33	ab	50.00	c
TmBt	57.55	c	75.00	c	TmBt	69.75	b	66.66	b
Tm	69.50	abc	83.33	b	Tm	89.29	ab	66.66	b

*** Ex vivo assay carried out with apple tree fruits, inoculated with *C. nymphaeae* and *C. chrysophillum*, previously treated with Eugenol (Eug); Eugenyl acetate (EgAc); Isoeugenol (Ieg); Thymol (Tm); Thymol acetate (TmAc); Thymol butyrate (TmBt); Carvacrol (Cv); Carvacryl acetate (CvAc); Carvacryl butyrate (CvBt); Dimethyl solfoxide (DMSO) and Manzate 800© two concentrations (Mzeb 100 and Mzeb 200). ** Original data, without processing; means followed by different letters in the column differ significantly by the Tukey’s test at 5% probability. *** Experiments 1 and 2 follow the same methodology and were conducted at separate time points with 5 months apart.

**Table 4 plants-13-03196-t004:** Severity and Incidence of Glomerella leaf spot after inoculation of a mixture of *Colletotrichum nymphaeae* and *Colletotrichum chrysophillum* isolates.

Experiment 1 ***					Experiment 2 ***				
Treatment *	Severity (%)		Incidence (%)		Treatment *	Severity (%)		Incidence (%)	
	14DAI		14DAI			14DAI		14DAI	
Control	34.17	a **	91.66	a	Control	70.00	a	91.66	a
CvAc	6.00	c	8.33	d	CvAc	0.05	b	8.33	c
EgAc	7.47	c	33.33	b	EgAc	0.00	b	0.00	d
TmAc	8.39	c	41.66	b	TmAc	2.00	b	25.00	b
CvBt	0.00	d	0.00	e	CvBt	0.34	b	16.66	b
Cv	0.12	d	8.33	d	Cv	0.08	b	8.33	d
Eug	0.12	d	8.33	d	Eug	1.02	b	66.66	a
Ieg	5.19	c	33.33	b	Ieg	0.51	b	41.66	a
Mzeb 100	0.00	d	0.00	e	Mzeb 100	0.05	b	16.66	b
Mzeb 200	13.08	b	16.66	c	Mzeb 200	0.00	b	0.00	d
TmBt	1.67	c	16.66	c	TmBt	0.05	b	16.66	b
Tm	15.32	b	41.66	b	Tm	0.00	b	0.00	d

* Ex vivo assay carried out with apple leves, inoculated with *C. nymphaeae* and *C. chrysophillum*, previously treated with Eugenol (Eug); Eugenyl acetate (EgAc); Isoeugenol (Ieg); Thymol (Tm); Thymol acetate (TmAc); Thymol butyrate (TmBt); Carvacrol (Cv); Carvacryl acetate (CvAc); Carvacryl butyrate (CvBt); Dimethyl solfoxide (DMSO) and Manzate 800© two concentrations (Mzeb 100 and Mzeb 200). ** Original data, without processing; means followed by different letters in the column differ significantly by the Tukey’s test at 5% probability. *** Experiments 1 and 2 follow the same methodology and were conducted at separate time points with 3 days apart.

## Data Availability

Data will be made available on request.

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
