# Peer review of "Eugenol, Isoeugenol, Thymol, Carvacrol, and Ester Derivatives as an Ecofriendly Option to Control Glomerella Leaf Spot and Bitter Rot on Apple"

_plants, 2024, doi:10.3390/plants13223196_

Round 1
Reviewer 1 Report
Comments and Suggestions for Authors The paper entitled “Eugenol, isoeugenol, thymol, carvacrol, and ester derivatives as anecofriendly option to control Glomerella leaf spot and Bitter rot on apple”
describes the study of some chemical derivatives of phenolic compounds
as an alternative route to control apple severe diseases instead of fungicides.
The paper seems to be interesting but some points should be clarified.
At first, the structure of the Ms should be corrected: the sections Methodology,
Results and Discussion contain in part the similar information that in fact is duplicated.
The section Methodology could be transformed to Experimental part (and contain
the data for equipment, reagents, compounds, procedures of chemical synthesis and
biological assays). If not, the section Methodology should present only the chosen
methodology (for ex, why exactly these phenolic compounds were taken, why these
routs for chemical derivatization was used etc), but not the details of chemical
procedures. The sections Results and Discussion could be combined in one section.
Figure 2: please revise the legend for compounds (lines 138-139): one can see,
that R1 for all 4 series is similar (CH3, (CH2)2CH3 and C6H6), so it could be
shown one time. What about the stability of the synthesized compounds?
Since they are the esters that could easily be hydrolyzed during the application
as antifungals please make the comments on this possible problem
The reference list: since the authors use the numbering of the references,
exactly this numbering should be used through the text of Ms.
Author Response
Dear Editor and Reviewers,
We sincerely appreciate the reviewers' contributions to improving our article. Below are all comments from each reviewer along with our responses. Changes are highlighted in the manuscript for easy identification. We have done our best to address the reviewers' suggestions. Should there be any further requests, please feel free to contact us.
The Authors
Comment 1)At first, the structure of the Ms should be corrected: the sections Methodology,
Results and Discussion contain in part the similar information that in fact is duplicated.
R: Thank you for your evaluation. We have reviewed and adjusted the document as requested, aiming to minimize repetitions, such as in the sentence mentioning that compound purification was performed using column chromatography.
Comment 2) The section Methodology could be transformed to Experimental part (and contain
the data for equipment, reagents, compounds, procedures of chemical synthesis and
biological assays). If not, the section Methodology should present only the chosen
methodology (for ex, why exactly these phenolic compounds were taken, why these
routs for chemical derivatization was used etc), but not the details of chemical
procedures.
R: Thank you very much for the suggestion. Modifications were made throughout Section 2, the Experimental part, to address Comment 2. These modifications are highlighted in Section 2 of the manuscript. The Experimental part now includes data and brands of the equipment used, with improved descriptions. Most justifications for the choice of compounds are provided throughout the text and are based on previous work by the same group, which involved a large selection of essential oils from medicinal plants and their major compounds. The initial work was previously published by Rozwalka et al., 2020, and this reference is included in the manuscript.
Comment 3) The sections Results and Discussion could be combined in one section.
R: Thank you for the suggestion, but at this moment we believe that the article is better understood with separate sections following the model that has been used in the Journal and in agreement with other reviewers of this manuscript. If it is mandatory that the sections be united in the editor's opinion, we can commit to doing so.
Articles with similar subject
https://www.mdpi.com/2223-7747/13/21/3077
https://www.mdpi.com/2223-7747/13/21/3076
4)Figure 2: please revise the legend for compounds (lines 138-139): one can see,
that R1 for all 4 series is similar (CH3, (CH2)2CH3 and C6H6), so it could be
shown one time.
R: Thank you for the suggestion. We have prepared an updated version of this figure, now included as Figure 2 in the manuscript.
5) What about the stability of the synthesized compounds?
Since they are the esters that could easily be hydrolyzed during the application
as antifungals please make the comments on this possible problem.
R: It is well known that ester hydrolysis is favored in strongly acidic or basic environments. However, during the experiments, the solutions and dispersions used did not exhibit pH levels in these extreme conditions. Thus, no decrease in the fungicidal activity of the esterified derivatives was observed throughout the tests. Additionally, in the case that hydrolysis does occur, control studies on the evaluated fungi have shown good activity for the individual phenolic components, which would be the products of potential hydrolysis. "Similarly, other studies (https://doi.org/10.1002/ps.4579; https://doi.org/10.1016/j.indcrop.2022.114855; https://doi.org/10.1016/j.phytol.2023.08.011) have synthesized and assessed ester derivatives for their biological activity across aqueous and non-aqueous media, with no evidence of hydrolysis detected throughout the experimental conditions.
We included in the discussion (highlighted in the text, section 5, line 557):
“A possible limitation in the experiment could be the sensitivity of esters to hydrolysis under extreme pH conditions, potentially compromising the fungicidal activity of the esterified derivatives. However, throughout the study, this did not occur. For future field studies, it is recommended to monitor pH levels to ensure product efficacy, especially if used in mixtures. Additionally, even if partial hydrolysis of the esters were to occur, the phenolic components resulting from hydrolysis exhibit strong antifungal activity.”
Comment 6) The reference list: since the authors use the numbering of the references,
exactly this numbering should be used through the text of Ms.
R: ok this was done.
We sincerely appreciate the time you’ve taken to review our manuscript and for offering valuable suggestions to enhance it. Thank you.
Reviewer 2 Report
Comments and Suggestions for Authors
Manuscript entitled “Eugenol, isoeugenol, thymol, carvacrol, and ester derivatives as an ecofriendly option to control Glomerella leaf spot and Bitter rot on apple”. This manuscript assessed the antifungal activity of eugenol, isoeugenol, thymol, carvacrol, and some of their ester derivatives. The manuscript provides significance for further control Glomerella leaf spot and bitter rot of apple. Minor points should be addressed before it can be accepted.
1. Line 168-169. The isolation of Ca32 and Col33 should be added.
2. Line 279. The calculation of disease severity and incidence should be added.
3. Table 2, 3, 4. The difference between Experiment 1 and 2 should be label in the Table captions.
Author Response
Dear Editor and Reviewers,
We sincerely appreciate the reviewers' contributions to improving our article. Below are all comments from each reviewer along with our responses. Changes are highlighted in the manuscript for easy identification. We have done our best to address the reviewers' suggestions. Should there be any further requests, please feel free to contact us.
The Authors
Comment 1) Line 168-169. The isolation of Ca32 and Col33 should be added.
R: Thank you very much for the suggestion. We have included three separate paragraphs to explain each methodology of the biological tests, with specific variations.
R: The fungal isolates Colletotrichum nymphaeae (isolate Ca32) and Colletotrichum chrysophillum (isolate Col33) from the LEMID-UFPR collection, originally isolated from GLS lesions on leaves, were transferred to potato dextrose agar (PDA) medium.
This was included in the section 2.5 and the isolates was always the same for next sections
Specific conditions for each kind of experiment were described in each section (2.5, 2.6 and 2.7)
Comment 2) Line 279. The calculation of disease severity and incidence should be added.
R: Thank you very much for the suggestion. We rewrite the sentence (highlighted in section 2.7 line 276).
“The incidence of symptomatic leaves was estimated for each treatment by the ratio between the number of leaves with GLS symptoms and the total leaves assessed. The GLS severity on leaves was estimated by calculating the average percentage of diseased area considering all leaves of each treatment with a standard area diagram set developed for this disease (Moreira et al. 2019). For fruit essays the incidence (number of diseased fruits in total fruit assessed) was quantified every day for 10 days. Also, the average of ABR lesion diameter was measure at 10 days after the inoculation (end of the experiment)”
We replace this sentence:
It was also calculated the average incidence and the average ABR lesion diameter 10 DAI and the average GLS severity and incidence 14 DAI for each treatment.
Comment 3) Table 2, 3, 4. The difference between Experiment 1 and 2 should be label in the Table captions.
R: Thank you very much for the suggestion. It was included in the legend part:
***Experiments 1 and 2 follow the same methodology and were conducted at separate time points with 30 days apart.
or
***Experiments 1 and 2 follow the same methodology and were conducted at separate time points with 5 months apart
We sincerely appreciate the time you’ve taken to review our manuscript and for offering valuable suggestions to enhance it. Thank you.
Reviewer 3 Report
Comments and Suggestions for Authors
This study aimed to assess the antifungal activity of eugenol, isoeugenol, thymol, carvacrol, and some of their ester derivatives in search of alternatives to inhibit the agents' diseases affecting apples: Glomerella Leaf Spot (GLS) and Bitter Rot (BR), respectively, Colletotrichum nymphaea and Colletotrichum chrysophillum.
The replacement of toxic pesticides with essential oil can be an option for biological control in apple orchards. So, the timeliness of the research is relevant, and the results can be considered a promising alternative in plant disease control.
I found the paper to be well written, without the problems of scientific expression.
The analysis methods are detailed, and the experimental design is correct.
The results are processed statistically, presented in tables and graphically,
The discussions are detailed, clearly expressed, and scientific, following similar research.
The bibliographic resources are relevant to recent data.
Minor observation
What do the values in bold in the Tables represent? To be specified in the Tables footnote.
The bibliography must be written in the order mentioned in the manuscript
Author Response
Dear Editor and Reviewers,
We sincerely appreciate the reviewers' contributions to improving our article. Below are all comments from each reviewer along with our responses. Changes are highlighted in the manuscript for easy identification. We have done our best to address the reviewers' suggestions. Should there be any further requests, please feel free to contact us.
The Authors
Comment 1) What do the values in bold in the Tables represent? To be specified in the Tables footnote.
R: Thank you very much for the suggestion. We remove all bold values. We did that just to facilitate the discussion during the writing
Comment 2) The bibliography must be written in the order mentioned in the manuscript
R: Thank you. Done.
We sincerely appreciate the time you’ve taken to review our manuscript and for offering valuable suggestions to enhance it. Thank you.